# Heading towards a dead end: The role of *DND1* in germ line differentiation of human iPSCs

Eva M. Mall[1,2], Aaron Lecanda🔵[1], Hannes C. A. Drexler[1], Erez Raz[3], Hans R. Schöler[1], Stefan Schlatt🔵[2]*

**1** Max Planck Institute for Molecular Biomedicine, Münster, Germany, **2** Centre of Reproductive Medicine and Andrology, Münster, Germany, **3** Institute of Cell Biology, ZMBE, Münster, Germany

* stefan.schlatt@ukmuenster.de

**Data Availability Statement:** All relevant data are within the paper and its Supporting Information files.

## Abstract

The *DND microRNA-mediated repression inhibitor 1* (*DND1*) is a conserved RNA binding protein (RBP) that plays important roles in survival and fate maintenance of primordial germ cells (PGCs) and in the development of the male germline in zebrafish and mice. Dead end was shown to be expressed in human pluripotent stem cells (PSCs), PGCs and spermatogonia, but little is known about its specific role concerning pluripotency and human germline development. Here we use CRISPR/Cas mediated knockout and PGC-like cell (PGCLC) differentiation in human iPSCs to determine if *DND1* (1) plays a role in maintaining pluripotency and (2) in specification of PGCLCs. We generated several clonal lines carrying biallelic loss of function mutations and analysed their differentiation potential towards PGCLCs and their gene expression on RNA and protein levels via RNA sequencing and mass spectrometry. The generated knockout iPSCs showed no differences in pluripotency gene expression, proliferation, or trilineage differentiation potential, but yielded reduced numbers of PGCLCs as compared with their parental iPSCs. RNAseq analysis of mutated PGCLCs revealed that the overall gene expression remains like non-mutated PGCLCs. However, reduced expression of genes associated with PGC differentiation and maintenance (e.g., *NANOS3*, *PRDM1*) was observed. Together, we show that *DND1* iPSCs maintain their pluripotency but exhibit a reduced differentiation to PGCLCs. This versatile model will allow further analysis of the specific mechanisms by which *DND1* influences PGC differentiation and maintenance.

## 1. Introduction

The *DND microRNA-mediated repression inhibitor 1* (*DND1*) is the human homolog of dead end (*dnd*), a conserved RNA binding protein (RBP) that plays an important role in survival and maintenance of primordial germ cells (PGCs) and the development of the male germline [1–3].

The gene was first discovered in zebrafish, where the protein is localized to perinuclear germ granules. Depleting *dnd* protein via anti-sense morpholinos oligonucleotides leads to

**Funding:** This work was financially supported by the Deutsche Forschungsgemeinschaft, Clinical Research Unit 326—'Male Germ Cells: from Genes to Function' (CRU326, SCHO 340/8-1 and SCHL 394/15-1) and by pilot project 6 of the CRU326. The funders had no role in study design, data collection and analysis, decision to publish, or preparation of the manuscript.

**Competing interests:** The authors have declared that no competing interests exist.

infertility as specified PGCs are lost upon migration due to apoptosis or transdifferentiation [4, 5]. *Dnd* is not necessary for the formation of PGCs in zebrafish [4, 5] or mouse [6, 7], but it is important for protecting PGCs from somatic differentiation cues while the cells migrate through the early developing embryo. Dnd protects the germ cell fate by inhibiting somatic differentiation, thereby maintaining the latent pluripotency of PGCs [4]. Upon arrival at the gonad, DND1 is important for downregulation of pluripotency factors and controls entry into the male germline program. It directly interacts with NANOS2 and the CCR4-NOT complex to initiate male differentiation [2, 6, 8–10]. At the molecular level, DND1 was shown to function via the interaction with RNAs, thereby fulfilling a variety of biological functions depending on target gene and developmental stage. It binds the 3'UTRs of target transcripts to stabilise the expression by antagonising miRNA mediated repression [11] or destabilizes mRNAs by direct recruitment of the CCR4-NOT deadenylase complex [6]. Its target spectrum includes signalling pathways regulating apoptosis [12], cell cycle [13], epigenetic modulations [14], and pluripotency [6, 9].

The Ter mutation in murine DND1 –introducing a premature stop codon that disrupts RNA binding–is associated with male infertility and with a sharp increase in teratoma formation when introduced in mice of the 129/sv background [7]. This emphasizes its role in maintaining the germ cell fate and in regulating the latent pluripotency of PGCs [1, 15].

In humans, *DND1* is expressed in migratory PGCs and iPSC-derived PGCLCs [16, 17]. Additionally, it is expressed in breast cancer cells, with elevated expression being associated with reduced malignancy and longer median survival, indicating a role as tumour suppressor by facilitating apoptosis [18]. Zhu and colleagues showed via over-expression of a DOX-inducible HA-tagged *DND1* in ESCs that the protein interacts with pluripotency factors (OCT4, SOX2, NANOG, LIN28), cell cycle regulators (TP53, LAT2) and apoptotic factors (BCLX, BAX) [19]. This raises the question, whether DND1 plays a role in maintaining pluripotency in PSCs.

Pluripotent stem cells are a valuable tool for developmental biology, as they possess the ability to differentiate *in vitro* towards different cell types of the body, including germ cells [20]. The development of protocols to differentiate human PSCs towards primordial germ cell like cells (PGCLCs) *in vitro* enabled for the first time the systematic study of germ cell development in man and showed that those first steps are orchestrated by a set of transcription factors that is different from mice [16, 21, 22]. Several subsequent studies demonstrated that the transcriptional profile of the generated cells is like that of migratory PGCs *in vivo*. Therefore, the developmental program of primordial germ cells can be to some extent recapitulated *in vitro* [23, 24]. Combining this differentiation with targeted genome editing enables to further elucidate the role of genes associated with infertility in early steps of differentiation. The CRISPR/Cas9 system is a versatile tool for genome editing in pluripotent stem cells [25]. Since its development in 2012 [26, 27], it became an important tool to manipulate the genome of pluripotent stem cells and is indispensable for disease modelling and drug discovery [28].

Here, we aimed at analysing the role of DND1 in human pluripotency and in the context of early germ cell differentiation. We generated DND1 knockouts in two different iPSC lines. We then analysed those lines for effects on gene expression patterns at RNA and protein levels and on the ability of the manipulated cells differentiating towards PGCLCs.

## 2. Methods

### 2.1 Pluripotent stem cell culture

Human iPSCs were derived from CD34$^+$ core blood stem cells (CB hiPSCs) or from testicular somatic cells (TS hiPSCs). Informed consent was obtained, and the investigation was approved

by the Ethics Committee of the University Hospital Münster. Cells were reprogrammed with lentiviral vectors encoding the human cDNA of OCT4, SOX2, KLF4 and c-MYC under the control of the SFFV promoter. Reprogramming and characterisation were described previously [29–31].

Human iPSCs were cultured in StemFlex™ medium (Thermo Fisher Scientific, Massachusetts, USA; #A3349401) in cell culture plates coated with Matrigel® (Corning, Massachusetts, USA; #354230) and maintained at 37˚C with 5% CO2 in a humified atmosphere. Medium was changed every day and cells were passaged every 3 days by singularization with TrypLE Express (Gibco, Thermo Fisher Scientific, Massachusetts, USA; #12604–013). 100,000 cells/ml were seeded and 10 µM Y27632 (ROCK inhibitor; Tebu-Bio, Germany; #21910–2301) were provided in the culture medium for 24 h to allow survival of single cells.

## 2.2 CRISPR/Cas9 mediated knockout and single cell expansion

For the CRISPR/Cas9 mediated knockout, we modified the published protocol of the Xhang laboratory [32]. Briefly, four sgRNAs targeting DND1 exon 1 or 2 were designed using a CRISPR design platform (http://www.e-crisp.org/E-CRISP/) (see Table 1). sgRNA oligonucleotides were cloned downstream of the human U6 promoter of a modified CRISPR/Cas9 expression plasmid from our group that allows for selection of successfully transfected cells via GFP expression (X330A-GFPT2APuro plasmid is available at Addgene #124202) [33]. Introduction of guide RNAs was analysed by end-point PCR using hU6 primer and the reverse guide as second primer. Positive clones were verified by sequencing.

Human iPSCs were transfected using Lipofectamine™ Stem Reagent (Thermo Fisher Scientific, Massachusetts, USA; STEM00008) according to the manufacturer's protocol. In brief, 200,000 cells were seeded per one well of a 24 well plate one day prior. On the day of lipofection, medium was changed 1–2 hours prior to the start to increase viability. To introduce the knockout, hiPSCs were transfected with a combination of two different sgRNA bearing plasmids to introduce a large deletion using 2 µl Lipofectamine™ and a total of 1 µg of plasmid DNA (0.5 µg per plasmid) in OptiMem™ (Gibco, Thermo Fisher Scientific, Massachusetts, USA; #11058021) medium with 2x RevitaCell™ (Thermo Fisher Scientific, Massachusetts, USA; #A16445-01). After 4 hours, cells were fed with an equal amount of StemFlex medium, and medium was changed to Stemflex without RevitaCell™ the next day. 24–48 hours after transfection (depending on the survival of the cells), hiPSCs were prone to single cell expansion using a protocol adapted from [34]. In short, hiPSCs were pre-treated with StemFlex + 1x RevitaCell™ for 1 hour prior to dissociation. For sorting, cells were resuspended in FB containing DAPI (5 ng/ml). Gating was adapted to sort for single and live cells. GFP⁺ cells were directly sorted on 96 well plates coated with Geltrax® (Thermo Fisher Scientific,

**Table 1. Guide RNA sequences.**

| Name | Target sequence | Sequence for gRNA cloning | Strand |
|------|------|------|------|
| DND1_gRNA-1 | GCCCGTTCACCTGCACCAGG NGG | CACCGCCCGTTCACCTGCACCAGG | plus |
| | | AAACCCTGGTGCAGGTGAACGGGC | |
| DND1_gRNA-2 | GACAGGCATCCGCCTGGTGC NGG | CACCGACAGGCATCCGCCTGGTGC | minus |
| | | AAACGCACCAGGCGGATGCCTGTC | |
| DND1_gRNA-3 | GGCTGGGACTACCGTACCTG NGG | CACCGGCTGGGACTACCGTACCTG | plus |
| | | AAACCAGGTACGGTAGTCCCAGCC | |
| DND1_gRNA-4 | GCTTACCTCACAATCCCGCT NGG | CACCGCTTACCTCACAATCCCGCT | plus |
| | | AAACAGCGGGATTGTGAGGTAAGC | |

Massachusetts, USA; #A1413302) with one cell per well and in StemFlex medium supplemented 1x RevitaCell™. Cells were fed once after 3 days with StemFlex + 1x RevitaCell™, then medium was changed every 3–4 days until colonies were ready to be harvested (usually after 12–14 days). For expansion, colonies were detached using Versene solution (Gibco, Thermo Fisher Scientific, Massachusetts, USA; #15040066) and re-seeded on 24 well plates (one colony per well). After reaching 60–80% confluency, cells were expanded to 6 well plates and part of the cells was used for DNA extraction. Clones carrying deletions in both alleles were identified by PCR and sequencing of the genomic PCR products (primers used for genotyping are listed in Table 2). Stocks were created from clonal lines with biallelic large deletions. Those stocks were tested to be free of mycoplasma. These cells were used for a maximum of 10 passages.

## 2.3 Differentiation of pluripotent stem cells

To analyse the differentiation potential of DND1 KO hiPSCs, the StemMACS™ Trilineage Differentiation Kit was used (Miltenyi Biotec, Germany; #130-115-660) according to the manufacturer's protocol. Differentiation efficiency was determined via flow cytometry. Therefore, cells were fixed and stained with the FoxP3 Staining Buffer kit for intracellular staining (Miltenyi Biotec, Germany; #130-093-142) with the following markers: OCT4, NANOG for pluripotency; CD140b for mesoderm, SOX17 for endoderm, SOX2 for ectoderm (antibodies are listed in Table 3). Undifferentiated cells and staining with isotype matched antibodies were used as controls.

For induction of PGCLCs, human iPSCs were subjected to a two-step differentiation approach, modified from previously published protocols [16, 22, 35]. For pre-induction, 200,000 cells/ml were seeded on Matrigel® (Corning, Massachusetts, USA; # 354230) coated plates in pre-induction medium: 50 ng/ml Activin A (Thermo Fisher Scientific, Massachusetts, USA; #PHC9563), and 3 µM CHIR99021 (Biomol, Germany; #Cay13122-10) in GK20: GMEM (Gibco, Thermo Fisher Scientific, Massachusetts, USA; #21710–025), 20% KnockOut™ serum replacement (Gibco, Thermo Fisher Scientific, Massachusetts, USA; #10828–028), 10 mM sodium pyruvate (Sigma-Aldrich, Germany; S8636-100ml), 1X non-essential amino acids (NEAA, Sigma-Aldrich, Germany; M7145-100ml). Y27632 (Tebu-Bio, Germany; # 21910–2301) was provided in a concentration of 10 µM for 24 h. After 48 h, primitive streak/mesodermal intermediate cells (referred to as incipient mesoderm-like or iMeLCs) were detached and aggregated with PGCLC differentiation medium either in 96-well of clear round bottom ultra-low attachment microplates (Corning®, MA, USA, #7007) or via the spin EB technique with AggreWell™400 microwells (Stemcell™ Technologies, Germany; #34415). PGCLC differentiation medium was GK20 supplemented with 200 ng/ml BMP4 (R&D Systems, Germany; #314-BP-01M), 1000 U/ml hLIF (Merk Millipore, Germany; #LIF1050), 50 ng/ml hEGF (Peprotec, Germany; #315-09-1000), 100 ng/ml mouse SCF (Thermo Fisher Scientific, Massachusetts, USA; #PMC2113L), and 10 nM Y27632 (Tebu-Bio, Germany; #21910–2301). In 96-well plates, 10,000 cells per aggregate were seeded in 80 µl medium per well and fed after 48 hours with another 80 µl medium without Y27632. The spin EB technique was performed according to the manufacture's protocol. In brief, plates were pre-treated with AggreWell™ Rinsing Solution (Stemcell™ Technologies, Germany; #07010), then 1,200,000 cells were seeded

**Table 2. Primer for genotyping and sequencing.**

| Target | Forward 5'-3' | Reverse 5'-3' |
|---|---|---|
| hU6 | GAGGGCCTATTTCCCATGATT | |
| pUC/M13 | CGCCAGGGTTTTCCCAGTCACGAC | CAGGAAACAGCTATGAC |
| DND1-DNA | AAGGTCATCATCAGGCGGAA | GTTATAAAGAGGGTACGAGGGGG |

**Table 3. Antibodies used for flow cytometry.**

| Antigen | Clone | Conjugation | Isotype | Specificity | Catalogue # | Supplier |
|---------|-------|-------------|---------|-------------|-------------|----------|
| **CD38** | HIT2 | APC | Mouse IgG1, κ | anti-human | 303510 | Biolegends |
| **TNAP** | B4-78 | PE | Mouse BALB/c IgG1, κ | anti-human | 561433 | BD Pharmingen™ |
| **CD140b** | REA363 | PE | Recombinant human IgG1 | anti-human | 130-123-995 | Miltenyi Biotec |
| **CD184 / CXCR4** | REA649 | APC | Recombinant human IgG1 | anti-human | 130-120-778 | Miltenyi Biotec |
| **SOX17** | REA701 | PE | Recombinant human IgG1 | anti-human | 130-111-148 | Miltenyi Biotec |
| **PAX6** | O18-1330 | Alexa Fluor® 488 | Mouse IgG2a, κ | anti-human | BD561664 | BD Pharmingen™ |
| **SOX2** | 245610 | Alexa Fluor® 647 | Mouse IgG2a | anti-human | BD560294 | BD Pharmingen™ |
| **Oct3/4** | REA622 | APC | Recombinant human IgG1 | anti-human, anti-mouse | 130-123-318 | Miltenyi Biotec |
| **NANOG** | REA314 | PE | Recombinant human IgG1 | anti-human | 130-117-526 | Miltenyi Biotec |
| **REA Control** | REA293 | PE | Recombinant human IgG1 | anti-human | 130-113-450 | Miltenyi Biotec |
| | | APC | | | 130-113-446 | |

(1,000 cells per aggregate) in differentiation medium and centrifuged for 5 min at 100 xg. After 48 h, cells were transferred to 6 cm low-adhesion plates and fed with differentiation medium without Y27632. PGCLCs were isolated by fluorescence-activated cell sorting (FACS) after 9–10 days of differentiation. Aggregates were pre-digested with 2 mg/ml collagenase IV and 2 mg/ml DNase I (in DMEM) for 15–25 min and were pipetted every 3–5 min to facilitate detachment into small clumps. For singularization, aggregates were incubated with TrypLE Express (Gibco, Thermo Fisher Scientific, Massachusetts, USA; #12604–013) for 5–10 min at 37˚C and passed through a pre-separation filter (Miltenyi Biotec, Germany, #130-041-407). Cells were stained for expression of CD38 and TNAP, if not stated otherwise (antibodies are listed in Table 3). Antibodies were diluted 1:50 in FACS buffer (FB: PBS with 3% fetal calve serum). Cells were incubated on ice for 30 min and subsequently washed 3 times with FB. Afterwards, cells were resuspended in FB containing DAPI (5 ng/ml). DAPI+ (dead) cells and remaining cell duplicates or cluster were excluded from sorting. Cells were sorted using FAC-SAria (BD Bioscience, New Jersey, USA) and collected in 1.5 ml low-adhesion reaction tubes and were frozen in -80˚C for subsequent RNA and protein isolation.

To estimate the size of the aggregates produced in 96-well plates, pictures of 10–12 aggregates randomly chosen from the plate were taken at 100x magnification. The aggregate area was measured and calculated applying the ImageJ software.

## 2.4 RT-qPCR

Gene expression of undifferentiated hiPSCs was analysed using quantitative real-time PCR (list of primers in Table 4). Total RNA was isolated from frozen cells using the NucleoSpin® Mini RNA isolation kit (Macherey-Nagel, Germany; #740955.250) according to the manufacturer's protocol. Concentration and purity of RNA were determined using the NanoDrop® 1000.

For reverse transcription, 0.5 µg of total RNA was transcribed in a total volume of 25 µl with 1X M-MLV reaction buffer, 0.2 mM oligo dT primer, 2 mM dNTP mix, and 40 U M-MLV reverse transcriptase (Affymetrix, USA). The reagents were incubated for 60 min at 42˚C followed by heat inactivation for 10 min at 75˚C. Resulting cDNA was stored at -20˚C. RT-qPCR was performed in 384-well plates using the QuantStudio™ 5 real-time PCR machine. 3 ng of RNA were added to a 10 µl reaction containing 1X of the iTaq SyBR® green supermix (Bio-Rad, Germany; #172–5120) and 0.125 µM of respective forward and reverse primers diluted in ddH$_2$O. Each sample was run in technical triplicates and samples without reverse transcriptase were used as negative control. Resulting data were analysed with the comparative

**Table 4. Primer sequences for RT-qPCR.**

| Target | Forward | Reverse | Source |
|---|---|---|---|
| ACTB | ATAGCAACGTACATGGCTGG | CACCTTCTACAATGAGCTGC | (Irie et al., 2015) |
| BLIMP1 | AAACCAAAGCATCACGTTGACA | GGATGGATGGTGAGAGAAGCAA | (Sasaki et al., 2015) |
| DND1 | TGCTGGGACAGGGACCTATG | ACGGCCATGGAAGATCACTG | (Irie et al., 2015) |
| DNMT3B | TAACTGGAGCCACGACGTAAC | GCATCCGTCATCTTTCAGCCTA | (Sasaki et al., 2015) |
| EOMES | AAGGGGAGAGTTTCATCATCCC | GGCGCAAGAAGAGGATGAAATAG | (Yokobayashi et al., 2017) |
| GAPDH | CGCTTCGCTCTCTGCTCCTCCTGT | GGTGACCAGGCGCCCAATACGA | (Irie et al., 2015) |
| NANOG | AGAGGTCTCGTATTTGCTGCAT | AAACACTCGGTGAAATCAGGGT | (Sasaki et al., 2015) |
| NANOS3 | TGGCAAGGGAAGAGCTGAAATC | TTATTGAGGGCTGACTGGATGC | (Sasaki et al., 2015) |
| OCT4 | CTGTCTCCGTCACCACTCTG | AAACCCTGGCACAAACTCCA | (Sasaki et al., 2015) |
| PRDM14 | TATCATACTGTGCACTTGGCAGAA | AGCAACTGGGACTACAGGTTTGT | (Sasaki et al., 2015) |
| SOX17 | TTCGTGTGCAAGCCTGAGAT | TAATATACCGCGGAGCTGGC | (Sasaki et al., 2015) |
| SOX2 | TGAATCAGTCTGCCGAGAATCC | TCTCAAACTGTGCATAATGGAGT | (Sasaki et al., 2015) |
| STELLA | AAGCCCAAAGTCAGTGAGATGA | GCTATAGCCCAACTACCTAATGC | (Sasaki et al., 2015) |
| T | AGCCAAAGACAATCAGCAGAAA | CACAAAAGGAGGGGCTTCACTA | (Sasaki et al., 2015) |
| TFAP2C | ATTAAGAGGATGCTGGGCTCTG | CACTGTACTGCACACTCACCTT | (Sasaki et al., 2015) |

Ct (ddCt) method. Briefly, ct values as mean of technical triplicates for each gene were normalised to the internal controls ACTB and GAPDH, and further to the respective calibrator — H9 hESC. Data is displayed as mean ± standard deviation (SD) unless stated otherwise.

## 2.5 Sample preparation for MS analysis

FAC-sorted parent cells ($2 \times 10^6$/replicate, 3 replicates per line) were prepared for bottom-up mass spectrometry analysis using an approach including in-solution digests and desalting, followed by high pH reversed phase chromatography fractionation using micro-columns (High pH RP Fractionation Kit, Pierce). Briefly, sorted cells were pelleted by centrifugation and the cell pellets lysed in 200μl hot lysis buffer (6M GdnHCl, 10mM TCEP, 40mM CAA, 25mM Tris-HCl pH8,5). Cell lysates were heated to 95°C for another 10min to complete denaturation and then centrifuged to remove cellular debris. Samples were cooled on ice, LysC was added directly to the supernatants (1:50) and incubated for 2 h at 37°C. Ammonium bicarbonate buffer (50mM) was then added to reduce the GdnHCl concentration to 1M and digestion extended overnight after the addition of trypsin (1:50). Digestion was stopped by the addition of TFA to a final concentration of 1% and digests were desalted using C18 cartridges (Empore 2M, Sigma). Desalted samples were then lyophilized prior to fractionation on high pH reversed phase microcolumns according to the manufacturer's instructions. Peptides in the resulting 8 fractions were aggregated into 4 pools, lyophilized and stored on stage tips prior to MS analysis.

Peptide samples in 0.1% formic acid were measured on a hybrid TIMS-quadrupole time of flight mass spectrometer (timsTOF pro) coupled to a nanoElute UHPLC system via a Captive Spray ion source (Bruker, Bremen, Germany). Peptides were separated on the nanoElute within 90 min with a linear gradient from 3 to 35% buffer B (Buffer A: 0.1% formic acid; Buffer B: 0.1% formic acid in acetonitrile) on a self-packed C18 reverse phase capillary column with pulled emitter tip (nanoseparations; 360 μm OD x 75 μm ID × 250 mm; Reprosil pur C18-aq, 1.9μm, Dr. Maisch) using a constant flow of 300 nl/min. At the end of the gradient, the column was flushed with 90% B before re-equilibration at starting conditions. MS and MS/MS spectra were recorded in positive mode from m/z 100 to 1700 Da, using the PASEF scan mode. Each duty cycle consisted of 1 TIMS-MS and an average of 10 PASEF MS/MS frames, each one

containing multiple MS/MS spectra, which resulted in a total cycle time of 1.1 s. To exclude the majority of singly charged ions with low m/z for PASEF precursor selection, a polygon filtering was applied to the m/z over ion mobility area. For the 90 min runs, target intensity was set to of 20.000 cts/s and an ion mobility range (1/K0) of 0.6–1.6 Vs/cm2 was used. Data were acquired with a 100 ms ramp time. The Bruker Hystar / oTOF Control software was used to control the whole LC-MS system and record the data (version 3.2, Bruker Daltonics, Bremen; Germany).

## 2.6 MS data analysis and label free quantification

MS files were processed using the MaxQuant computional platform (version 1.6.14.0; [36]). Identification of peptides and proteins was enabled by the built-in Andromeda search engine by querying the concatenated forward and reverse human Uniprot database (UP000005640_9606.fasta; version from 04/2019) including common lab contaminants. Default values of MaxQuant remained unchanged. Trypsin was selected as protease allowing up to two missed cleavages, and the peptide search was limited to a minimum length of 7 amino acids and a maximum mass of 4600 Da. Oxidation of methionine and protein N-terminal acetylation were set at variable modifications and carbamidomethylations of cysteine as fixed modification. For peptide and protein identifications a minimum false discovery rate (FDR) of 1% was required. The match between runs option was enabled setting a retention time matching window of 0.7 min and a $1/K_0$ matching window of 0.05 V·s/cm$^2$.

Relative label free quantification using the MaxQuant LFQ algorithm was based on the measurements of 3 biological replicates for each sample. Data processing was performed using Perseus (version 1.6.14.0) [37]. First, reverse and contaminant hits as well as proteins, that were identified by a single modified peptide only, were eliminated from the list of identified protein groups. Proteins eventually included for further analysis had to be identified with at least 1 unique peptide. LFQ intensity values were log2 transformed and filtered. At least 2 valid values were present in at least one of the experimental groups. Still missing values (NaN) were replaced by imputation (downshift 1.8, width 0.3), simulating signals of low abundant proteins within the distribution of measured values. PCA analysis was also performed within the Perseus. To identify in a supervised manner the sets of proteins that significantly distinguish the different parent cell lines, an ANOVA multiple sample test with a permutation-based FDR of 0.05 to correct for multiple hypothesis testing was performed. The heat map was generated on significantly changed proteins only following normalization by Z–scoring.

The mass spectrometry data generated and analysed in this article have been deposited to the ProteomeXchange Consortium via the PRIDE partner repository [38] with the dataset identifier PXD023802.

## 2.7 Sample preparation for RNAseq analysis

Total RNA was isolated from 1 x10^6 to 2 x10^6 hiPSCs or iMeLCs using the NucleoSpin® Mini RNA isolation kit (Macherey-Nagel, Germany; #740955.250) and from 1.5–6 x10x^4 sorted PGCLCs (CD38$^+$/TNAP$^+$) according to the manufacturer's protocol. Concentration and purity of RNA were determined using the NanoDrop® 1000. mRNA was purified using the NEBNext Poly(A) mRNA magnetic isolation module (New England Biolabs® GmbH, Germany; #E7490S) and the library was prepared with the NEBNext Ultra II RNA Library Prep Kit for Illumina (New England Biolabs® GmbH, Germany; #E7770S) according to the manufacturer's protocol. The sequencing was performed with the Illumina NextSeq 550 platform using the NextSeq 500/550 High Output Kit (v2.5, 150 Cycles, Illumina, California USA, #20024907).

## 2.8 RNAseq data analysis

The samples were first demultiplexed using the Illumina ™ software bcl2fastq v2.19.0. The quality of the resulting fastq files was then assessed with FastQC v0.11.8 (https://www.bioinformatics. babraham.ac.uk/projects/fastqc/).

The samples were then pseudo aligned to the human genome using SALMON v1.4.0 [39], followed by a quantification of gene expression levels performed by the same program. The version of the human genome used as reference was GRCh38.p13 with the gene and feature annotations from GENCODE v36.

The secondary analysis was then performed with the statistical software R v4.0.3 (https:// www.r-project.org/). The gene count tables were first imported with the package tximeta v1.8.4 [40]. As a quality filter, only genes that had at least 5 reads in 3 or more samples were kept for further processing.

The Variance Stabilizing Transformation was applied over the read counts to estimate a dispersion trend among the samples, with which PCA plots were generated.

Differential gene expression was calculated using DEseq2 v1.30.1 [41]. Unless particularly specified otherwise in the results section, genes were considered as differentially expressed (DE) if they had an absolute value of $\log_2$ fold change $\geq 1.5$, and a Benjamini-Hochberg false discovery rate $\leq 5\%$

Heatmaps of the expression of selected DE genes were created using the package pheatmap v1.0.12 for the values obtained after applying the Variance Stabilizing Transformation on the read counts.

The data was uploaded to the GEO repository under accession number GSE174464.

**2.8.1 Addition of external datasets.** Fastq files form the publication of Irie et al. 2015 were downloaded from the Sequence Read Archive (SRA, https://www.ncbi.nlm.nih.gov/ sra) repository by using the corresponding accession IDs and the fasterq-dump command from the SRA toolkit v2.10.5. We used the same programs and reference as for our own samples to align and process the external samples. Data merging and comparisons were performed in R with the same packages and tools we used to process our own samples.

## 2.9 Gene ontology (GO) term analysis

Functional enrichment analysis on the input gene list of differentially expressed genes found via mass spectrometry and RNA sequencing were performed using the g:GOSt analyser (g:Profiler version e102_eg49_p15_7a9b4d6) with the standard g:SCS significance threshold and the user threshold set to 0.05 [42].

## 2.10 Statistical analysis

For statistical analysis of aggregate size and cell numbers after differentiation, pairwise t-tests with BH adjustment were performed. We considered p<0.05 as statistically significant.

## 3. Results

### 3.1 Generation and characterisation of DND1-KO hiPSCs

Large deletions in DND1 were introduced in two iPSC lines using the CRISPR/Cas9 system. One line derived from core blood (CB) and the second line from testicular somatic (TS) cells. Clonal lines with biallelic large deletions were identified via PCR and mutations were verified via sequencing (Fig 1A and S1 Raw images and S2 File). All analysed deletions introduced frame shifts and premature stop codons. For each iPSC line, three DND1$^{-/-}$

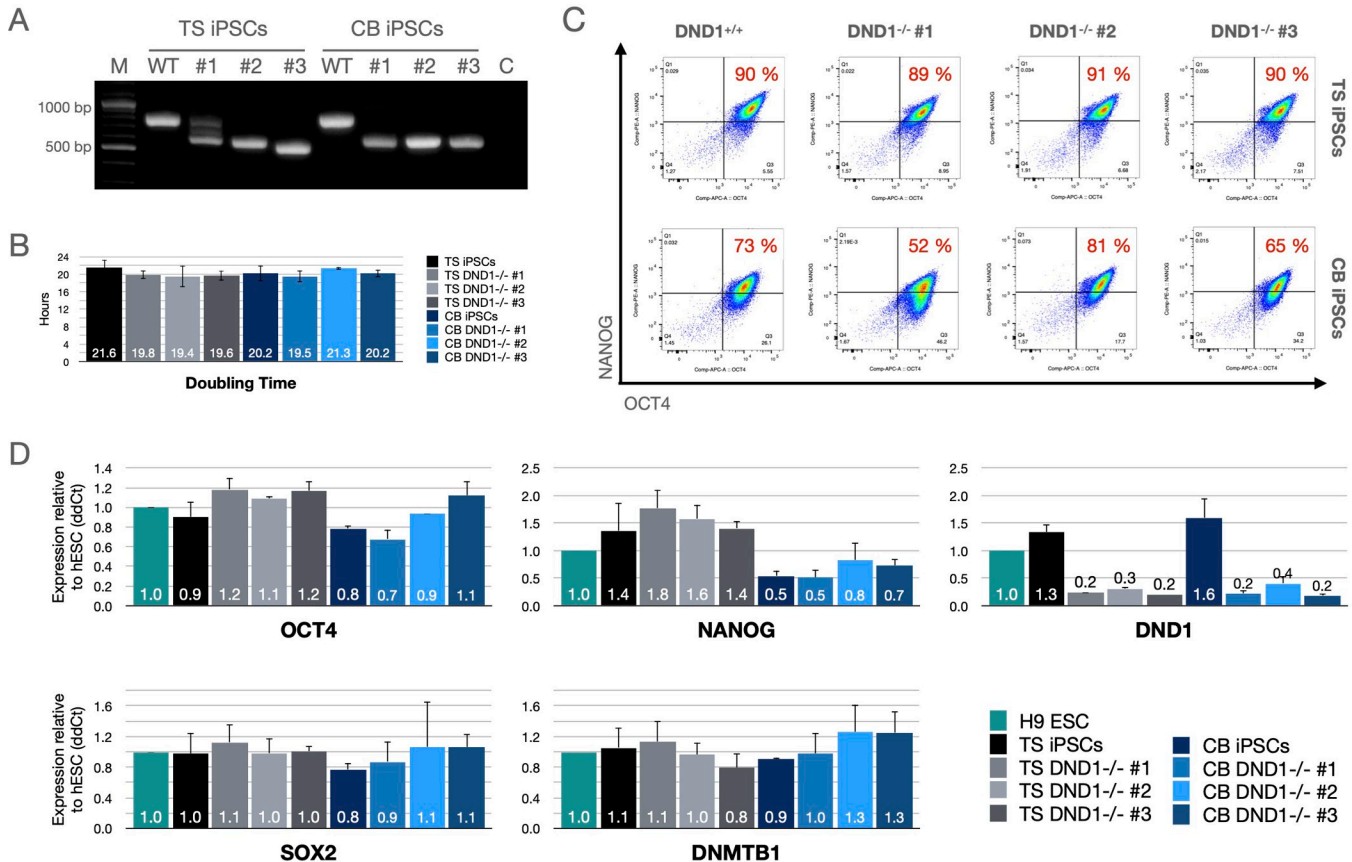

**Fig 1. Characterisation of DND1$^{-/-}$ hiPSCs.** (A) End-point PCR showing presence of large deletions in DND1 genomic locus. (B) All lines exhibited similar doubling times of about 20 hours. Doubling times counted from 4 independent passages per line. (C) About 90% of TS hiPSCs were double positive for OCT4 and NANOG, irrespective of DND1 status. CB hiPSCs showed reduced NANOG expression with 52–81% double positive cells. (D) RT-qPCR showed similar expression of OCT4, SOX2, and DNMTB1 in all lines compared to H9 hESCs. CB hiPSCs showed about 50% reduction of NANOG expression, irrespective of DND1 status. DND1 is expressed in low levels in hESCs and DND1$^{+/+}$ hiPSCs and mRNA levels are strongly reduced in DND1$^{-/-}$ hiPSCs. The data were normalised to two internal controls (ACTB and GAPDH) and are shown as fold change to H9 ESCs. N = 3 biological replicates.

clones were used for further analyses (Table 5). All clones exhibited normal colony morphology and growth rate (Fig 1B). To ensure that pluripotency was not disrupted by introduction of DND1 mutations and clonal expansion, we examined the expression of pluripotency factors via flow cytometry (Fig 1C) and RT-qPCR (Fig 1D). To demonstrate their differentiation potential, all lines were shown to be responsive to a standardised trilineage differentiation protocol. All WT and DND1 KO lines differentiated towards the three lineages to a similar extend (S1 Fig in S1 File).

**Table 5. DND1 knockout iPSCs lines for analysis.**

| Clone | iPSC line | Transfected with | Deletion | Stop codon after |
|---|---|---|---|---|
| TS DND1$^{-/-}$ #1 | TS iPSCs | gRNA -1 + -4 | 30 bp and 215 bp | 12 AA and 7 AA |
| TS DND1$^{-/-}$ #2 | | gRNA -2 + -4 | 216 bp | 110 AA |
| TS DND1$^{-/-}$ #3 | | gRNA -3 + -4 | 258 bp | 47 AA |
| CB DND1$^{-/-}$ #1 | CB iPSCs | gRNA -1 + -4 | 215 bp | 7 AA |
| CB DND1$^{-/-}$ #2 | | gRNA -1 + -4 | 215 bp | 7 AA |
| CB DND1$^{-/-}$ #3 | | gRNA -2 + -4 | 216 bp | 110 AA |

## 3.2 Mass spectrometric analyses

To further analyse potential differences in global protein expression of the different iPSCs, we analysed the non-differentiated iPSCs by mass spectrometry. Principal component analysis showed that the CB and TS iPSCs formed separate clusters, with DND1$^{+/+}$ and $^{-/-}$ iPSCs clustering together, indicating that different characteristics arise from the iPSC line and are not associated with the DND1 knockout (Fig 2A). For each line, three replicates were analysed, whereby one of the replicates of CB WT and CB #3 clustered separate for unknown reasons. Closer examination of the differentially expressed proteins and the corresponding gene ontology classification revealed 5 main clusters: Cluster 1 (higher expression in CB iPSCs) was enriched for structural constituents of the cytoskeleton, cadherin binding and carboxylic acid metabolic processes; Cluster 2 (higher expression in CB #2 and #3) was enriched for GO terms associated with ATP binding, cellular metabolic processes and multicellular organism development; Cluster 3 (higher expression in CB #1) contained components of APP-FOXO4 complex; Cluster 4 (higher expression in TS iPSCs) was enriched for phosphoric ester hydrolase activity; Cluster 5 (higher expression in TS iPSCs) was enriched for cytoskeletal and cell adhesion protein binding, cell-extracellular matrix interactions, and anatomical structure morphogenesis (Fig 2B and 2C and S1 Data). No differences in pluripotency-related proteins were detected. Taken together, differences among the cell lines seem to arise from parental iPSC lines and include primarily general house-keeping genes.

## 3.3 Differentiation capacity

To analyse the effect of DND1 on the ability of iPSCs to contribute to the germ line, we differentiated all lines towards PGCLCs using the spin EB method for aggregate formation. We analysed PGCLC formation using CD38 and TNAP as markers. TS iPSCs generated similar ratios of double positive PGCLCs (10–20% of total cells). The absolute number of PGCLCs was reduced in TS DND1$^{-/-}$ by about 30–50% compared to TS DND1$^{+/+}$ (Fig 3A, upper row). CB

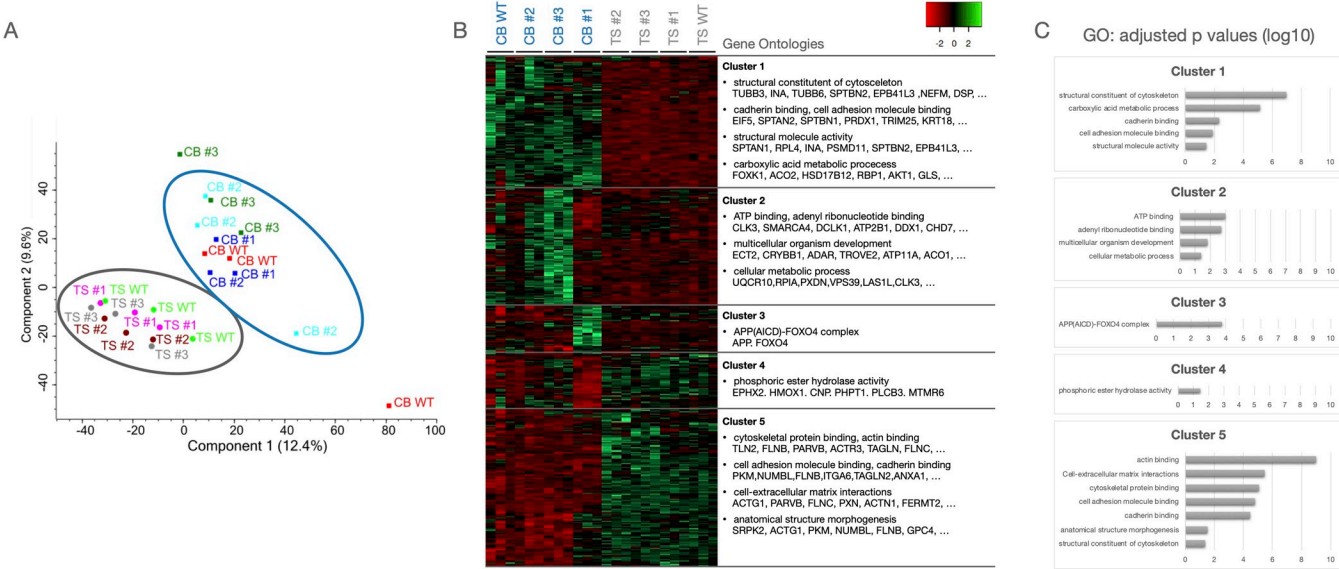

**Fig 2. Proteome analysis of the different iPSCs.** (A) Principal component analysis shows that the TS and CB lines cluster together, respectively, irrespective of their DND1 status. (B) Analysis of the differentially expressed proteins revealed 5 clusters. Highly represented gene ontology clusters and exemplary proteins are depicted. (C) Bar graphs showing the adjusted p values for the mentioned GO terms per cluster (log10 scale).

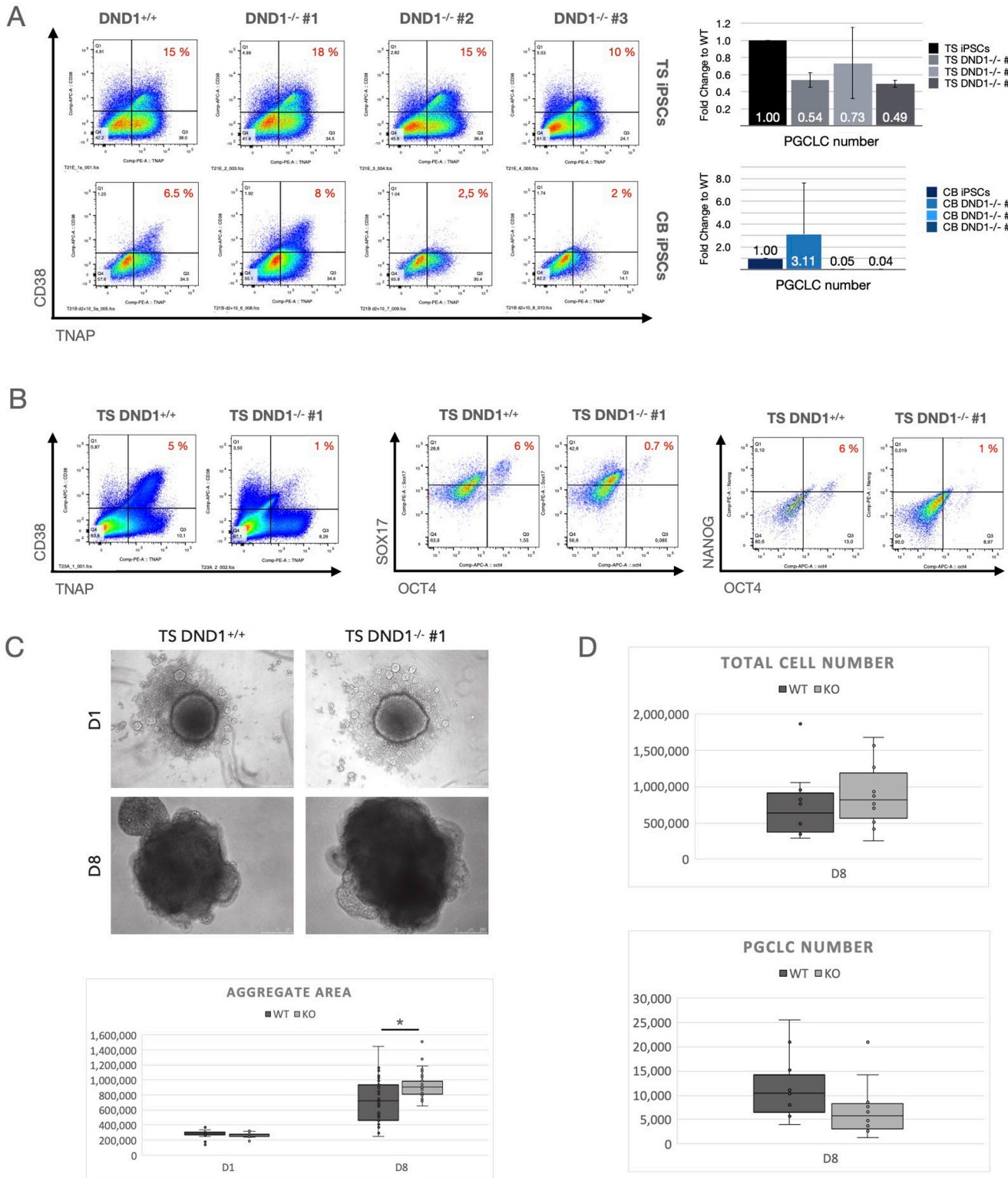

**Fig 3. Differentiation towards PGCLCs.** (A) All lines were differentiated using the spin EB method and sorted at D8 for CD38/TNAP double positive PGCLCs. Left: Flow cytometry results shown as dot plots, right: bar graphs with total PGCLC numbers derived. TS DND1^{-/-} hiPSCs resulted in similar ratios but reduced total numbers compared to DND1^{+/+}. CB hiPSCs differentiated less efficiently, and CB DND1^{-/-} #2 & #3 died during PGCLC induction, resulting in low total cell numbers and very few PGCLCs. CB DND1^{-/-} #1 showed efficient differentiation resulting in increased numbers of PGCLCs compared to CB DND1^{+/+} (n = 3). (B) We verified the results with TS DND1^{+/+} and TS DND1^{-/-} #1 after aggregation in 96 well plates using staining against

CD38/TNAP, OCT4/SOX17 and OCT4/NANOG. All three marker combinations resulted in reduced ratios and absolute numbers of PGCLCs in TS DND1$^{-/-}$ #1 compared to wildtype iPSCs. (C) We measured aggregate sizes (visible area) in D1 and D8 aggregates and found that DND1$^{-/-}$ #1 iPSCs aggregates were bigger compared to the control (n = 3 experiments, with 10 aggregates measured per cell line and experiment). Difference on D8 is significant with * p< 0.01 tested with pairwise t-tests. (D) When we analysed the total cell numbers derived per experiment, TS DND1$^{-/-}$ #1 showed higher total cell numbers while PGCLCs were reduced (n = 10 experiments). Differences in total and PGCLC number did not reach statistical significance between WT and KO cells (pairwise t-test with p<0.05).

iPSCs differentiated less efficiently (2–8% double positive cells in DND1$^{+/+}$). Cells belonging to DND1$^{-/-}$ #1 maintained proper aggregate formation and generated PGCLCs more efficiently compared to non-mutated cells. DND1$^{-/-}$ #2 and #3 formed proper aggregates at D1, but over time those aggregates became smaller and at D5-8, only a small number of viable cells were recovered. Almost no PGCLCs were detected (Fig 3A, lower row).

We then re-examined the effect with TS DND1$^{+/+}$ and TS DND1$^{-/-}$ #1 iPSCs using 96 well low adhesion plates for aggregate formation. The total number of cells recovered was higher compared to the Spin EB method, while the number of PGCLCs remained similar. Therefore, the difference in PGCLC recovery was also manifested in a reduced ratio of CD38/TNAP positive cells (Fig 3B, left). We confirmed this change by flow cytometry, monitoring the additional PGCLC marker combinations OCT4/SOX17 and OCT4/NANOG (Fig 3B, middle and right). We observed that DND1$^{-/-}$ aggregates tended to be larger on D8 (Fig 3C and S2 Fig in S1 File) and the total number of cells increased (Fig 3D, upper), indicating that reduced PGCLC numbers were not resulting from a general loss of cells, but was specific for PGCLCs (Fig 3D, lower).

## 3.4 RNAseq analyses

Next, we analysed the effect on gene expression after loss of DND1 by RNA sequencing. First, we applied a variance stabilizing transformation to visualize similarities and differences among different cell fractions. WT and KO cluster together in the PCA plot with differences resulting mainly from the cell type as PGCLCs from iPSCs and iMeLCs cluster separately (Fig 4A).

To identify differentially expressed genes, we performed multiple comparison analysis between WT and KO cell using a false discovery rate of 5% and the threshold for log$_2$ fold change of 1.5. Two samples (iMeLCs and PGCLCs of KO #3) had to be excluded due to low read count. For iPSCs, 12 genes showed differential expression (6 higher and 6 lower) and in iMeLCs only one gene revealed higher and 5 genes a lower expression, indicating that loss of DND1 had no biologically relevant effect on the gene expression in iPSCs and iMeLCs. In PGCLCs, 4 genes showed a higher expression and 23 genes exhibited reduced expression in KO compared to WT cells (Fig 4B). The list of differentially expressed genes is presented in S2 Data. Within the 23 downregulated genes, *NANOS3* and *PRDM1* were most noticeable, as both are important regulators of PGC specification and maintenance (Fig 4B).

We compared our data set to the published data set from Irie and colleagues [16]. Our iPSCs and iMeLCs clustered together with their conventionally cultured ESCs, but our PGCLCs (both WT and KO) clustered apart from PGCLCs they reported. We sorted our PGCLCs after 9–10 days in aggregates, while Irie and colleagues characterised cells after 4 days in aggregates. We considered a fold change of 4 and a FDR of 1% and analysed the 1000 most abundant genes. Under these criteria, 279 genes showed a lower and 721 genes a higher expression in the Irie-generated PGCLCs as compared to our PGCLCs. The GO term analysis indicated that the genes with higher expression in Irie-PGCLCs are relevant for extracellular matrix organization and associated developmental processes. Elevation of mRNA levels in our PGCLCs were enriched in GO terms for negative epigenetic regulation of gene expression and transcription. No differences were detected in the expression of genes associated with PGC

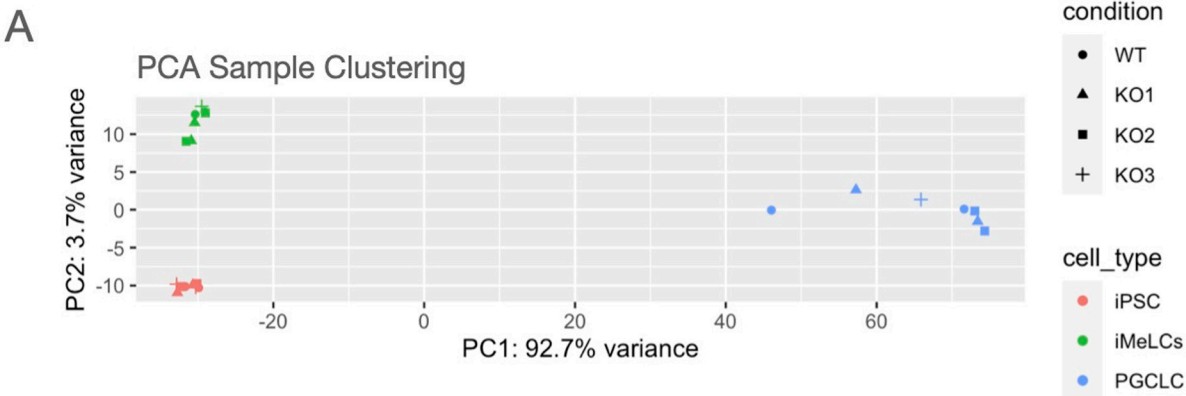

**Fig 4. RNA expression analysis.** (A) PCA clustering showed that DND1 KO lines clustered together with DND1 WT lines. Main differences are associated with the differentiation status, with PGCLCs clustering distinct from iPSCs and iMeLCs. (B) Heatmap showing differentially expressed genes in PGCLCs. DND1 KO PGCLCs show decreased expression in known PGC differentiation regulators as PRDM1 and NANOS3 (cluster 1).

differentiation, such as *NANOS3* or *TFAP2C*. This confirms that our cells are indeed PGCLCs but may possess a different state of epigenetic regulation eventually due to the longer culture period (see S3 Fig in S1 File, differentially expressed genes, and GO terms are presented in S3 Data).

## 4. Discussion

Here, we present the generation and characterisation of DND1 knockout iPSCs. Our data indicate that loss of DND1 does not disrupt pluripotency in iPSCs, but interferes with es the maintenance of PGCLCs after differentiation.

Bi-allelic loss of function mutations were established in two different iPSC lines. Loss of *DND1* did neither alter cell doubling time, differentiation to the three germ layers nor expression on RNA and proteins in undifferentiated cells. Endpoints were established via RT-qPCR, mass spectrometry and bulk RNA sequencing. Expression of DOX-inducible DND1 in human embryonic stem cells identified different transcripts that interact with DND1, including pluripotency factors, cell cycle regulators and apoptotic factors [19]. DND1 is expressed in pluripotent stem cells at relatively low levels, indicating that it does not play an important role in maintaining pluripotency in iPSCs and that loss of DND1 does not have a significant impact on the overall gene expression in iPSCs.

We did observe clear differences between the two parental iPSC lines, irrespective of the *DND1* activity. While the functional pluripotency assays indicate similar potential, CB iPSCs showed reduced *NANOG* expression at RNA and protein levels. Mass spectrometry indicated differences in the expression of cytoskeletal components. Variability among different iPSC lines was reported in several contexts and may arise from genetic background, reprogramming method, or epigenetic memory [43, 44]. It was indeed shown that clonal variation influences PGCLC differentiation efficiency [45]. Differences in the abundance of cytoskeletal proteins were described to decrease the differentiation potency to derive mesodermal and endodermal tissue [46]. As PGCLC differentiation is initiated over a meso-endodermal precursor state (iMeLCs), this may also have implications for the efficiency of PGCLC differentiation. The parental iPSCs showed no differential expression of pluripotency factors during proliferation. There were also no differentially expressed proteins between the $DND1^{+/+}$ and $DND1^{-/-}$ TS hiPSCs, indicating that loss of *DND1* itself has no global effect on protein expression. However, we detected differences between $DND1^{+/+}$ and $DND1^{-/-}$ CB iPSCs: KO 2 and 3 exhibited higher expression in Cluster 2 that is enriched for GO terms linked to cellular metabolic processes, ATP binding and multicellular organism development. None of those proteins are directly associated with PGCLC differentiation, but changes in the energy metabolism might account for the strongly reduced survival of those lines during aggregate formation and PGCLC differentiation.

Although pluripotency was not altered in iPSCs by the loss of *DND1*, we observed reduced numbers of PGCLCs after differentiation. This indicates that the cells can undergo PGCLC differentiation but do so at reduced efficiency. Alternatively, PGCLCs may be lost after specification. RNAseq analysis confirmed that gene expression in our knockout PGCLCs differed only slightly from the wildtype. However, two important regulators of germline development and maintenance, *NANOS3 and PRDM1*, are expressed at lower levels. Both proteins are known to protect PGCLCs from somatic differentiation cues, so we assume that reduced expression will facilitate trans-differentiation. In zebrafish, *dnd* is important for maintaining the latent pluripotency program in PGCs and protects the cells from external cues while migrating through the developing embryo. Indeed, loss of *dnd* then leads to a trans-differentiation towards other somatic lineages [4]. The idea that the newly formed PGCLCs of our DND1 KO iPSCs trans-

differentiate towards other lineages and are not lost by apoptosis is supported by the observation that aggregate size and total cell numbers are increased in TS KO aggregates compared to the WT counterpart. This is in agreement with studies in zebrafish and mouse showing that DND is not needed for PGC specification, but is of upmost importance for maintenance and further development of the germline [1, 4, 6]. Our current results may also be explained by differentiation of iMeLCs to other cell types showing stronger mitotic expansion. More *in depth* analysis of the non-PGCLC fraction will be needed to characterize and identify all cell types within our aggregates.

To further elucidate when exactly PGCLCs are lost and to which cell types they potentially trans-differentiate, a more extensive in-depth analysis at different timepoints on single cell level would be required. Here, we analysed sorted PGCLCs towards the end of differentiation. Transdifferentiated cells were not captured. Comparing WT and KO using scRNAseq may thus provide a more detailed picture. As PGCLCs in zebrafish adopt to their surrounding tissue, we assume that the trans-differentiation in the *in vitro* culture will not be random but may reflect the various routes that gradients of BMP4 can induce.

*DND1* knockout iPSCs have strong potential for further studies and could be helpful to explore the specific role of *DND1* in mammalian germ cell development. As further specification of PGCs depends on signals from the somatic environment, it would be interesting to explore interaction with somatic cells. We have recently established a novel xeno-organoid approach that enables co-culture of human PGCLCs with rat somatic testicular tissue [47]. By integrating our *DND1* KO lines with this xeno-organoid model, we can analyse these effects in more detail. In addition, loss of *DND1* in zebrafish was associated with reduced migration [4]. PGCLCs show only limited migration *in vitro* [35], but this feature might only be observed when combining somatic and germ cells.

## Supporting information

**S1 Raw images. Raw image of gel data.**
(TIFF)

**S1 File.**
(DOCX)

**S2 File. Sequences of DND1 knockout iPSCs.**
(DOCX)

**S1 Data. Clusters of differentially expressed proteins detected in mass spectrometric analysis.**
(XLSX)

**S2 Data. List of differentially expressed genes between WT and KO iPSCs.**
(XLSX)

**S3 Data. List of differentially expressed genes between Mall and Irie data set.**
(XLSX)

## Acknowledgments

The authors want to thank Martin Stehling for excellent assistance with flow cytometric analysis and cell sorting and Christoph Börling for help with the statistical analysis.

## Author Contributions

**Conceptualization:** Eva M. Mall, Erez Raz, Hans R. Schöler, Stefan Schlatt.

**Data curation:** Aaron Lecanda, Hannes C. A. Drexler, Erez Raz.

**Formal analysis:** Aaron Lecanda, Hannes C. A. Drexler.

**Funding acquisition:** Erez Raz, Hans R. Schöler, Stefan Schlatt.

**Investigation:** Eva M. Mall, Aaron Lecanda, Erez Raz, Stefan Schlatt.

**Methodology:** Eva M. Mall, Hannes C. A. Drexler, Erez Raz, Hans R. Schöler, Stefan Schlatt.

**Project administration:** Hans R. Schöler, Stefan Schlatt.

**Resources:** Hans R. Schöler, Stefan Schlatt.

**Supervision:** Stefan Schlatt.

**Validation:** Eva M. Mall, Aaron Lecanda, Stefan Schlatt.

**Visualization:** Eva M. Mall, Hannes C. A. Drexler, Stefan Schlatt.

**Writing – original draft:** Eva M. Mall, Stefan Schlatt.

**Writing – review & editing:** Hannes C. A. Drexler, Erez Raz, Hans R. Schöler, Stefan Schlatt.

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
