## [Decision Letter · Decision Letter 0]

25 May 2021

PONE-D-21-14615

Heading towards a dead end: the role of DND1 in germ line differentiation of human iPSCs

PLOS ONE

Dear Dr. Schlatt,

Thank you for submitting your manuscript to PLOS ONE. After careful consideration, we feel that it has merit but does not fully meet PLOS ONE’s publication criteria as it currently stands. Therefore, we invite you to submit a revised version of the manuscript that addresses the points raised during the review process.

We look forward to receiving your revised manuscript.

Kind regards,

Wei Cui, Ph.D.

Academic Editor

PLOS ONE

Journal Requirements:

2.PLOS ONE now requires that authors provide the original uncropped and unadjusted images underlying all blot or gel results reported in a submission’s figures or Supporting Information files. This policy and the journal’s other requirements for blot/gel reporting and figure preparation are described in detail at https://journals.plos.org/plosone/s/figures#loc-blot-and-gel-reporting-requirements and https://journals.plos.org/plosone/s/figures#loc-preparing-figures-from-image-files. When you submit your revised manuscript, please ensure that your figures adhere fully to these guidelines and provide the original underlying images for all blot or gel data reported in your submission. See the following link for instructions on providing the original image data: https://journals.plos.org/plosone/s/figures#loc-original-images-for-blots-and-gels.

3.Thank you for stating the following in the Acknowledgments Section of your manuscript:

"This work was financially supported by the Deutsche Forschungsgemeinschaft, Clinical Research Unit

500 326—‘Male Germ Cells: from Genes to Function’ (CRU326, SCHO 340/8-1 and SCHL 394/15-1) and by

501 a pilot project funding by the CRU326 (pilot project 6)."

Additional Editor Comments:

While all three reviewers found this manuscript interesting, several major concerns were raised about figure legends, data in some figures, cell line/lineage markers, and manuscript writing. Please refer to reviewers' comments for details.

Reviewers' comments:

Reviewer's Responses to Questions

**Comments to the Author**

1. Is the manuscript technically sound, and do the data support the conclusions?

Reviewer #1: No

Reviewer #2: Partly

Reviewer #3: Partly

2. Has the statistical analysis been performed appropriately and rigorously? 

Reviewer #1: No

Reviewer #2: No

Reviewer #3: I Don't Know

3. Have the authors made all data underlying the findings in their manuscript fully available?

Reviewer #1: No

Reviewer #2: Yes

Reviewer #3: Yes

4. Is the manuscript presented in an intelligible fashion and written in standard English?

Reviewer #1: Yes

Reviewer #2: Yes

Reviewer #3: No

5. Review Comments to the Author

Reviewer #1: In this manuscript, the authors try to show functions of DND1, an evolutionally conserved genes crucial for germ cell development, in human primordial germ cell-like cell (PGCLC) differentiation from iPS cells (iPSC) by its knockout. Loss of DND1 did not result in significant changes of pluripotency and proteomic profiles as well as of global gene expression profiles in the two different iPSCs, while efficiency of PGCLC differentiation from the iPSCs and the expression of NANOS3 and PRDM1, which are important for PGC development, in the induced PGCLCs, were reduced by DND1 deficiency. Roles of DND1 in human PGC development are potentially interesting, but this study does not reach a definitive conclusion, because it is unreliable that only subtle reduction (~1.5 folds) of the two gene expression without additional changes of required gene expression affects PGCLC specification. In addition, data presentation and discussion are often inadequate as described below.

Specific comments:

1. Labeling for Fig.1~Fig.3 may be wrong. This reviewer made comments for the figures according to the figure number in figure legends in the text.

2. Fig.3A is not informative, which should be deleted, because DND1 KO did not influence efficiency of PGCLC differentiation from TS iPSCs, and did not result in consistent effects in CB iPSC in this method (Spin-EB). Instead, the authors should show the effect of DND1 KO by the low adhesion plate methods shown in Fig.3B-D in more detail, including the effects of DND1 KO in different TS DND1-/- cell lines (#2, #3) as well as CB DND1-/- lines. In Fig. 3C, D, statistical evaluation is necessary.

3. Fig. 1A; Sequences of the deleted regions should be shown.

4. Fig. 1B, D; Number of biological replicates should be shown.

5. Reduced expression of NANOS3 and PRDM1 in DND1 KO PGCLCs should be confirmed by RT-qPCR.

6. Supplementary Fig. S1; FACS profiles of the undifferentiated iPSCs as well as of the cells stained by the control antibodies should be shown.

7. Transcriptomes of PGCLCs from parent WT iPSCs in this study and by Irie et al. are substantially different as shown in Supplementary Fig. S3, and the authors discussed that difference of the culture period may cause the difference (line 411~). If they think so, they should repeat transcriptome analysis by using day 9~10 PGCLCs. Although the authors discussed that their cells were indeed PGCLCs due to the expression of a few marker genes without showing it (line 419), it is likely that the cells in this study may not be proper PGCLCs judged by the data presented in this manuscript. More definitive criteria and experimental evidence supporting that their cells are PGCLCs, are necessary.

8. The authors discussed a possibility that loss of DND1 cause trans-differentiation of PGCLCs, but it is too speculative without definitive experimental evidence (line 466~). It is more likely that iPSCs or iMeLCs differentiated from iPSCs may differentiate not to PGCs, but to different types of cells with active proliferation.

Reviewer #2: Mall et al. generated DND1-null human iPSC clones by CRISPR/Cas9 and tested their differentiation potencies. Authors claimed a significant reduction in efficiency of PGCLC production for the DND1-null iPSCs compared to the parental iPSCs whereas the potencies to generate the three germ layers seemed unaffected. Authors also argued that the DND1-null PGCLCs expressed reduced levels of genes associating PGC differentiation (e.g., NANOS3 or PRDM1).

The claimed specific importance of DND1 in PGCLC production from iPSCs is interesting, and the proposed requirement of DND1 in the PGC-like transcriptomal characteristics of PGCLCs seems significant. However, this Reviewer feels that more convincing data are necessary to publish such significant conclusions. Especially, the RNA-seq data shown as Figure 4B need more work to convincingly support the claims of Authors.

[Major Points]

1) Figure 2A. Authors are requested to explain why three datum points of the same sample show significant variations. See CB WT, CB #2, and CB #3 datum points that are located far from the other points.

1) The RNA-seq heatmap data (Figure 4B) show very strong difference between the two columns under “TS WT” – namely, the left column is mostly bright red whereas the right column is mostly black with only a few dark red genes. If the wild type-derived PGCLCs already show quite unstable transcriptomal characteristics, comparison of their transcriptomes with DND1-null PGCLCs would be difficult to interpret. Authors are requested to repeat PGCLC production from TS WT to confirm that the differential expression of the PGC markers shown in Figure 4 is not due to technical inconsistencies.

2) Figure 4B does not include CD38. Because PGCLCs were FACS-enriched using CD38 and TNAP, Authors are requested to show CD38 expression in Figure 4B to make sure that all cells presented in Figure 4B are indeed CD38+ PGCLCs. If CD38 expression is also very low in DND1-knockout cells than wild type cells, the identities of cells presented in this figure would be questionable.

3) Figure 4B shows only representative marker genes and/or differentially expressed genes. Authors are requested to show genes whose mRNA expression is not affected by DND1 knockdown. It is expected that many genes are not affected by DND1 knockdown; however, if this is the case, Authors need to evaluate viabilities of cells to exclude the possibility that the remarkable transcriptional contrasts shown in Figure 4B is not due to non-specific damages.

[Minor Points]

1) Figure numbers are incorrect. Figure 1 is actually Figure 2, Figure 2 is Figure 3, and Figure 3 is Figure 1. A figure with no figure number is Figure 4.

2) Figure S3 is not mentioned in the text. Perhaps Authors want to refer to it in the paragraph starting Line 409.

3) There are many typographical errors. For example,

Line 34: is ac conserved (is conserved?)

Line 47: NANOS2 and the CXCR4-NOT complex (CCR4-NOT?)

Line 56: stop codon that RNA binding– is associated (stop codon in the RNA binding domain?)

Line 67: This rases the question (raises the question?)

Line 79: in early steps of differentiation, The CRISPR/Cas9 (differentiation. The CRISPR/Cas9?)

Line 466: Both proteinsare as well (Both proteins are as well?)

4) Erez Raz is listed as a co-author on the title page, but he is acknowledged in the Acknowledgments section – “We want to thank Eres Raz for useful comments on the manuscript.” (Line 498).

5) The zebrafish gene “dead end” appears in the abstract and introduction (Lines 16 and 38). Authors may want to amend the text to make it clear that this is a name of gene.

6) Lines 421-422: (see Supplemental Fig. 2). Are Authors requesting to see a supplemental materials of a reference (Irie et al.) cited 12 lines above?

Reviewer #3: The work in this manuscript investigates the role of the conserved protein DND1 in pluripotency and germline development in human cord blood and testicular somatic cell derived iPSCs. The requirement for Dnd was examined using CRISPR/Cas9 to disrupt Dnd in two distinct human cell lines. The gene expression patterns of “wild-type” and Dnd deficient cell lines were characterized using RNA seq (to examine transcripts) and mass spec (to analyze proteins). Comparison of the RNA and protein expression profiles and Flow cytometry analysis and cell sorting revealed no major differences in pluripotency factors or ability to differentiate along various lineages in the absence of Dnd. Based on the data provided wild-type and mutant datasets clustered according to the cell type they were derived from. Because mutant aggregates were larger than those of wild-type and fewer PGCLCs were recovered in sorting assays, it is suggested that Dnd1 may be required to maintain PGCLC fate in this context, as it is in other animals. Sequencing of PGCLCs from wild-type and mutants and comparison to previously published datasets suggests that expression of differentiation genes was intact, but that extracellular matrix regulators were lower in mutants whereas negative regulators of epigenetic state were elevated in mutant PCGLCs. The data appear to be properly controlled, are overall clearly presented within each figure, and are consistent with the main conclusion that Dnd1 is not required for pluripotency or differentiation, but instead promotes maintenance of germline fate in this system as has been previously observed in animal models. This was clearly a lot of work, but enthusiasm for the manuscript is diminished because grammatical errors within the text, mismatch between the figures and legends, and in a few cases lack of detail in the legend detract from the clarity of the manuscript. The experiments overall are well described; however, the main conclusions are not consistently clearly stated. This may be in part due to the variability observed from line to line.

Specific comments:

1) The main figures and legends appear to be out of order. Figure legend 1 seems to match Figure 3, legend 2 matches Figure 1, Legend 3 matches Figure 2.

2) Were statistical analyses performed for the data presented in Figure 2 (legend 3)?

3) A more detailed legend is needed for Supplementary Fig. 3. It is not clear which dataset is being compared for Mall data – I this both the bulk sequencing of the aggregates and/or sequencing of the sorted PGCLCs? Also, it is not clear which data are from the wild-type cells and the mutant cells? or is this just wild-type? For the GO analysis it is not clear what was compared – the full dataset or specific clusters from the Irie dataset. Please clarify these points.

4) It is suggested that newly formed PGCLCs transdifferentiate toward other lineages and that trans-differentiation in the culture system is not random and may reflect a shift toward fates dependent on BMP4. Is there any hint from the sequencing data that this is the case?

5) It is stated that “DND1 is expressed in pluripotent stem cells at relatively low levels, so we assume that loss of function was compensated by other factors.” It is unclear why low levels of expression would suggest compensation – please clarify.

6) The mass spectrometry experiments are described in the methods and text and a link to the data was provided, but the data or a summary thereof do not appear to be included in the main or supplemental figures. It is unclear why was the mouse Uniprot database used for the MS analysis.

7) One of the authors, Erez Raz, is acknowledged and his name is misspelled in the Acknowledgements section (which has a typo in the title). It is thoughtful but unusual to acknowledge an author in this section as typically authorship itself acknowledges an individual’s contribution.

8) The author lists don’t match between the main paper and supplemental – Erez Raz is listed as an author on the main paper but not on the supplemental data file.

9) There are typos and grammatical errors throughout the manuscript. For example “extend” should be “extent”, “intercipient” should be “incipient”, “sells” should be “cells”, instead of “prone to” consider “subjected to”, articles are missing in several places, and some sentences are unclear (e.g. lines 84-86 on page 5, 146-147 on page 7, and there is a comma instead of a period on line 79 pg 4, words are missing in several sentences including page 3 line 56 where it appears “disrupts” is missing before “RNA binding”, pg2 line 27 appears to be missing “with”, on page 3 ,line 34 “is ac” should be “is a”, and page 4 line 76 has an extra “that”. This list is not all-inclusive, please proof-read carefully.

10) Table 2: is the reverse primer missing for hU6 or was only one primer used?

6. PLOS authors have the option to publish the peer review history of their article (what does this mean?). If published, this will include your full peer review and any attached files.

Reviewer #1: No

Reviewer #2: **Yes: **Toshi Shioda

Reviewer #3: No

---

## [Author Response · Author response to Decision Letter 0]

10 Aug 2021

Response to Reviewers

We are delighted that our manuscript has been reviewed by three specialists in the field who provided constructive criticisms on our manuscript. We respond to the invitation for submission of a revised version. We provide answers to all critical comments and have intensely re-analyzed our datasets. We are now showing all differentially expressed genes and the raw data as supplementary materials. We also reformatted and reformulated our statements and hope thereby to improve the validity of our study. Please find below a detailed response to all suggestions made by the reviewers.

Reviewer #1:

In this manuscript, the authors try to show functions of DND1, an evolutionally conserved genes crucial for germ cell development, in human primordial germ cell-like cell (PGCLC) differentiation from iPS cells (iPSC) by its knockout. Loss of DND1 did not result in significant changes of pluripotency and proteomic profiles as well as of global gene expression profiles in the two different iPSCs, while efficiency of PGCLC differentiation from the iPSCs and the expression of NANOS3 and PRDM1, which are important for PGC development, in the induced PGCLCs, were reduced by DND1 deficiency. Roles of DND1 in human PGC development are potentially interesting, but this study does not reach a definitive conclusion, because it is unreliable that only subtle reduction (~1.5 folds) of the two gene expression without additional changes of required gene expression affects PGCLC specification. In addition, data presentation and discussion are often inadequate as described below.

We agree that it is interesting and important to elucidate the role of DND1 in PGC development. We were intrigued to see PRDM1 and NANOS3 being affected. We were also surprised, as is the reviewer, to see no major changes in gene expression. However, all three reviewers agree that we performed a valid study with slightly surprising outcome. In the following, we deal with the specific comments point by point.

Specific comments:

1. Labeling for Fig.1~Fig.3 may be wrong. This reviewer made comments for the figures according to the figure number in figure legends in the text.

We apologize for this mistake and corrected the numbering.

2. Fig.3A is not informative, which should be deleted, because DND1 KO did not influence efficiency of PGCLC differentiation from TS iPSCs, and did not result in consistent effects in CB iPSC in this method (Spin-EB). Instead, the authors should show the effect of DND1 KO by the low adhesion plate methods shown in Fig.3B-D in more detail, including the effects of DND1 KO in different TS DND1-/- cell lines (#2, #3) as well as CB DND1-/- lines. In Fig. 3C, D, statistical evaluation is necessary.

It is noteworthy that both, SpinEB method and EB generation via low adhesion plates result in comparable differentiation towards PGCLCs. This is consistent with the methodological outcome previously shown by Mitsunaga and colleagues. (https://doi.org/10.1073/pnas.1707779114). 

As preliminary experiments (data not shown), we validated our strategy with various inhouse iPSC lines. This results in similar outcomes applying both methods. Albeit not reaching statistical significance, we observed in five of six lines a reduction of PGCLCs recovered from DND1-/- iPSCs. Following this clear trend, we decided to use the Spin EB method for the comparison of all lines in parallel. Due to loss of cells in CB DND1-/- lines, it was not possible to reliably measure the size of EBs using the low adhesion plate method. 

We added during revision the results of the statistical analysis for Fig. 3.

3. Fig. 1A; Sequences of the deleted regions should be shown.

We included the sequences in the supplementary information as text files and indicate it in the results section (Supplement 1)

4. Fig. 1B, D; Number of biological replicates should be shown.

The number of biological replicates was added to the figure legends.

5. Reduced expression of NANOS3 and PRDM1 in DND1 KO PGCLCs should be confirmed by RT-qPCR.

Several publications in recent years have shown a strong linear relationship between RT-pPCR and RNA sequencing. It was shown that RNA sequencing is quantitatively more reliable and more sensitive and must be considered superior in detecting low abundant transcripts. We therefore believe that RT-qPCR data would not improve the validity to our findings.

6. Supplementary Fig. S1; FACS profiles of the undifferentiated iPSCs as well as of the cells stained by the control antibodies should be shown.

We included controls (undifferentiated iPSCs for all lines and exemplary plots used for setting the gates for isotype matched control antibodies) in the supplementary file (Suppl. Fig. 1)

7. Transcriptomes of PGCLCs from parent WT iPSCs in this study and by Irie et al. are substantially different as shown in Supplementary Fig. S3, and the authors discussed that difference of the culture period may cause the difference (line 411~). If they think so, they should repeat transcriptome analysis by using day 9~10 PGCLCs. Although the authors discussed that their cells were indeed PGCLCs due to the expression of a few marker genes without showing it (line 419), it is likely that the cells in this study may not be proper PGCLCs judged by the data presented in this manuscript. More definitive criteria and experimental evidence supporting that their cells are PGCLCs, are necessary.

The co-expression of marker genes specific for PGCs and PGCLCs was previously described. Here we used two pairs of genes that are commonly used to identify PGCs and PGCLCs: CD38 & TNAP and OCT4 & SOX17 (see Fig.3). On the other hand, OCT4 and NANOG expression is rapidly lost in differentiated cell types, except for PGCLCs. We are therefore convinced that our marker panel can unequivocally prove the presence of PGC-like cells.

8. The authors discussed a possibility that loss of DND1 cause trans-differentiation of PGCLCs, but it is too speculative without definitive experimental evidence (line 466~). It is more likely that iPSCs or iMeLCs differentiated from iPSCs may differentiate not to PGCs, but to different types of cells with active proliferation.

We are thankful for this comment and discussed this view. The reviewer notes correctly that iMeLCs can directly differentiate to other cell types as is known from zebrafish. As our statements may have been misleading, we rephrased this passage and include the alternative option. 

Reviewer #2:

Mall et al. generated DND1-null human iPSC clones by CRISPR/Cas9 and tested their differentiation potencies. Authors claimed a significant reduction in efficiency of PGCLC production for the DND1-null iPSCs compared to the parental iPSCs whereas the potencies to generate the three germ layers seemed unaffected. Authors also argued that the DND1-null PGCLCs expressed reduced levels of genes associating PGC differentiation (e.g., NANOS3 or PRDM1).

The claimed specific importance of DND1 in PGCLC production from iPSCs is interesting, and the proposed requirement of DND1 in the PGC-like transcriptional characteristics of PGCLCs seems significant. However, this Reviewer feels that more convincing data are necessary to publish such significant conclusions. Especially, the RNA-seq data shown as Figure 4B need more work to convincingly support the claims of Authors.

We agree that the role of DND1 needs to be analyzed in more detail as it appears relevant for the maintenance of PGCLCs. We are sorry to not having described the evidence in sufficient detail. Our heatmap is focused on the differentially expressed genes. However, it is evident from the PCA clustering (Fig. 4A) that most genes are not affected. While only minor changes in global gene expression occur, it is interesting that NANOS3 and PRDM1 are differentially expressed indicating a common pathway. Future studies should focus on single cell analysis of differentiated cells present in the non-PGC fraction. We have modified and extended this this section in the revised discussion.

[Major Points]

1a) Figure 2A. Authors are requested to explain why three datum points of the same sample show significant variations. See CB WT, CB #2, and CB #3 datum points that are located far from the other points.

We assume that these differences arise from differential growth of samples prior to the analysis. We repeated the cluster analysis with and without those samples and found that they had no influence on the overall clusters of differentially expressed genes. 

1b) The RNA-seq heatmap data (Figure 4B) show very strong difference between the two columns under “TS WT” – namely, the left column is mostly bright red whereas the right column is mostly black with only a few dark red genes. If the wild type-derived PGCLCs already show quite unstable transcriptomal characteristics, comparison of their transcriptomes with DND1-null PGCLCs would be difficult to interpret. Authors are requested to repeat PGCLC production from TS WT to confirm that the differential expression of the PGC markers shown in Figure 4 is not due to technical inconsistencies.

Fig. 4 B shows a heatmap of genes differentially expressed in KO cells compared to WT cells. Statistically different values are detected at a log2 fold change of 1.5 and a false discovery rate of 5%. At this level, we see significant differences in WT control and between WT and KO cells. We performed this analysis to reveal, that the detected differences are not due to stochastic effects. 

The number of differentially expressed genes is low. The majority of detected transcripts are present in similar numbers as shown by PCA (Fig.4A). This is supporting the information that WT and KO PGCLCs belong to one cluster which is apart from other cell types.

2) Figure 4B does not include CD38. Because PGCLCs were FACS-enriched using CD38 and TNAP, Authors are requested to show CD38 expression in Figure 4B to make sure that all cells presented in Figure 4B are indeed CD38+ PGCLCs. If CD38 expression is also very low in DND1-knockout cells than wild type cells, the identities of cells presented in this figure would be questionable.

Fig. 4b includes only differentially expressed genes after sorting for CD38. Therefore, cells with substantially different (especially lower) expression were not present in the analyzed cell fraction. We apologize that with the first submission the link for the transcriptome repository was missing. All sequencing data are now uploaded to a public repository (To review the GEO accession GSE174464: Go to https://www.ncbi.nlm.nih.gov/geo/query/acc.cgi?acc=GSE174464 and enter token clydwicqzjyxnip into the box.). Thereby, the list of all detected mRNA transcripts, including those with no differential expression, is accessible to all readers.

3) Figure 4B shows only representative marker genes and/or differentially expressed genes. Authors are requested to show genes whose mRNA expression is not affected by DND1 knockdown. It is expected that many genes are not affected by DND1 knockdown; however, if this is the case, Authors need to evaluate viabilities of cells to exclude the possibility that the remarkable transcriptional contrasts shown in Figure 4B is not due to non-specific damages.

Sorting included selection for viability with Hoechst staining during FAC sorting. 

Results for genes that are not differentially expressed are found in the complete RNAseq data set, that we uploaded to a public repository (See our previous response for the link). Generally, the number of differentially expressed genes is relatively low, so most genes are not affected by the knockout, what is as well visible in the PCA plot, as +/+ and -/- cells cluster together.

[Minor Points]

1) Figure numbers are incorrect. Figure 1 is actually Figure 2, Figure 2 is Figure 3, and Figure 3 is Figure 1. A figure with no figure number is Figure 4.

Figure numbering was corrected

2) Figure S3 is not mentioned in the text. Perhaps Authors want to refer to it in the paragraph starting Line 409.

Thanks for pointing this out, we included it in the text.

3) There are many typographical errors. For example,

Line 34: is ac conserved (is conserved?)

Line 47: NANOS2 and the CXCR4-NOT complex (CCR4-NOT?)

Line 56: stop codon that RNA binding– is associated (stop codon in the RNA binding domain?)

Line 67: This rases the question (raises the question?)

Line 79: in early steps of differentiation, The CRISPR/Cas9 (differentiation. The CRISPR/Cas9?)

Line 466: Both proteins are as well (Both proteins are as well?)

We carefully proof-read again, thanks for pointing out the most crucial mistakes.

4) Erez Raz is listed as a co-author on the title page, but he is acknowledged in the Acknowledgments section – “We want to thank Eres Raz for useful comments on the manuscript.” (Line 498).

We corrected this mistake.

5) The zebrafish gene “dead end” appears in the abstract and introduction (Lines 16 and 38). Authors may want to amend the text to make it clear that this is a name of gene.

Done as requested.

6) Lines 421-422: (see Supplemental Fig. 2). Are Authors requesting to see a supplemental materials of a reference (Irie et al.) cited 12 lines above?

We referred here to our supplemental figure S3 and corrected the text.

Reviewer #3:

The work in this manuscript investigates the role of the conserved protein DND1 in pluripotency and germline development in human cord blood and testicular somatic cell derived iPSCs. The requirement for Dnd was examined using CRISPR/Cas9 to disrupt Dnd in two distinct human cell lines. The gene expression patterns of “wild-type” and Dnd deficient cell lines were characterized using RNA seq (to examine transcripts) and mass spec (to analyze proteins). Comparison of the RNA and protein expression profiles and Flow cytometry analysis and cell sorting revealed no major differences in pluripotency factors or ability to differentiate along various lineages in the absence of Dnd. Based on the data provided wild-type and mutant datasets clustered according to the cell type they were derived from. Because mutant aggregates were larger than those of wild-type and fewer PGCLCs were recovered in sorting assays, it is suggested that Dnd1 may be required to maintain PGCLC fate in this context, as it is in other animals. Sequencing of PGCLCs from wild-type and mutants and comparison to previously published datasets suggests that expression of differentiation genes was intact, but that extracellular matrix regulators were lower in mutants whereas negative regulators of epigenetic state were elevated in mutant PCGLCs. The data appear to be properly controlled, are overall clearly presented within each figure, and are consistent with the main conclusion that Dnd1 is not required for pluripotency or differentiation, but instead promotes maintenance of germline fate in this system as has been previously observed in animal models. This was clearly a lot of work, but enthusiasm for the manuscript is diminished because grammatical errors within the text, mismatch between the figures and legends, and in a few cases lack of detail in the legend detract from the clarity of the manuscript. The experiments overall are well described; however, the main conclusions are not consistently clearly stated. This may be in part due to the variability observed from line to line.

We are sorry for the reviewers impression that we have not taken sufficient care for language errors. We have now performed additional proofreading and hope to eliminate most concerns. We also checked that our legends are detailed and that our conclusions are consistent throughout the manuscript.

Specific comments:

1) The main figures and legends appear to be out of order. Figure legend 1 seems to match Figure 3, legend 2 matches Figure 1, Legend 3 matches Figure 2.

We are sorry for the confusion and corrected the mistake.

2) Were statistical analyses performed for the data presented in Figure 2 (legend 3)?

We performed additional data analysis and statistics and include the information on the results of the statistical analysis in the revised figure.

3) A more detailed legend is needed for Supplementary Fig. 3. It is not clear which dataset is being compared for Mall data – I this both the bulk sequencing of the aggregates and/or sequencing of the sorted PGCLCs? Also, it is not clear which data are from the wild-type cells and the mutant cells? or is this just wild-type? For the GO analysis it is not clear what was compared – the full dataset or specific clusters from the Irie dataset. Please clarify these points.

We include a more detailed description. We only compared our WT cells (iPSCs, iMeLCs, sorted PGCLCs) against ESCs and the sorted PGCLCs from the Irie data set. We generally did no bulk sequencing of aggregates. For the GO term analysis, we compared our PGCLCs against the PGCLCs from the Irie data set.

4) It is suggested that newly formed PGCLCs transdifferentiate toward other lineages and that trans-differentiation in the culture system is not random and may reflect a shift toward fates dependent on BMP4. Is there any hint from the sequencing data that this is the case?

We did not analyze the -non-PGCLC fraction. We speculate that human PGCLCs behave similar to zebrafish PGCs and will react to developmental cues in the surrounding microenvironment. As high levels of BMP4 are present in the differentiation medium, we assume that trans-differentiation is prompted by this potent mesodermal morphogen. As this appears slightly speculative, we changed the sentence and more carefully interpret our findings in the revised paper.

5) It is stated that “DND1 is expressed in pluripotent stem cells at relatively low levels, so we assume that loss of function was compensated by other factors.” It is unclear why low levels of expression would suggest compensation – please clarify.

We assume that DND1 plays a minor role in maintaining pluripotency in iPSCs and that therefore loss of DND1 does not have a significant impact on gene expression in this cell type. We agree that the text was misleading and changed the sentence. 

6) The mass spectrometry experiments are described in the methods and text and a link to the data was provided, but the data or a summary thereof do not appear to be included in the main or supplemental figures. It is unclear why was the mouse Uniprot database used for the MS analysis.

Human Uniprot database was used for comparison. We corrected this mistake in the methods section. The mass spectrometry results are shown in Fig.2 (proteome analysis). This may have overseen due to the mix-up of the figure labels. We corrected this mistake.

7) One of the authors, Erez Raz, is acknowledged and his name is misspelled in the Acknowledgements section (which has a typo in the title). It is thoughtful but unusual to acknowledge an author in this section as typically authorship itself acknowledges an individual’s contribution.

Corrected 

8) The author lists don’t match between the main paper and supplemental – Erez Raz is listed as an author on the main paper but not on the supplemental data file.

Apologies for this inconsistency, we corrected this mistake

9) There are typos and grammatical errors throughout the manuscript. For example “extend” should be “extent”, “intercipient” should be “incipient”, “sells” should be “cells”, instead of “prone to” consider “subjected to”, articles are missing in several places, and some sentences are unclear (e.g. lines 84-86 on page 5, 146-147 on page 7, and there is a comma instead of a period on line 79 pg 4, words are missing in several sentences including page 3 line 56 where it appears “disrupts” is missing before “RNA binding”, pg2 line 27 appears to be missing “with”, on page 3 ,line 34 “is ac” should be “is a”, and page 4 line 76 has an extra “that”. This list is not all-inclusive, please proof-read carefully.

Thanks for pointing out those mistakes, we corrected them and proof-read carefully again.

10) Table 2: is the reverse primer missing for hU6 or was only one primer used?

hU6 is used in combination with the corresponding reverse guide RNA to verify the presence of guides in correct orientation in the plasmid by end-point PCR, therefore only one primer was listed. We clarified this in the method section.

---

## [Editor Report · Decision Letter 1]

28 Sep 2021

Heading towards a dead end: the role of DND1 in germ line differentiation of human iPSCs

PONE-D-21-14615R1

Dear Dr. Schlatt,

We’re pleased to inform you that your manuscript has been judged scientifically suitable for publication and will be formally accepted for publication once it meets all outstanding technical requirements.

Kind regards,

Wei Cui, Ph.D.

Academic Editor

PLOS ONE

Additional Editor Comments (optional):

Questions and concerns have been addressed.

Authors need to double check their uploaded figures, which are duplicated with different numbers.
---

## [Editor Report · Acceptance letter]

8 Oct 2021

PONE-D-21-14615R1 

Heading towards a dead end: the role of *DND1* in germ line differentiation of human iPSCs 

Dear Dr. Schlatt:

I'm pleased to inform you that your manuscript has been deemed suitable for publication in PLOS ONE. Congratulations! Your manuscript is now with our production department. 

Kind regards, 

on behalf of

Prof. Wei Cui 

Academic Editor

PLOS ONE